# "If I don't take my treatment, I will die and who will take care of my child?": An investigation into an inclusive community-led approach to addressing the barriers to HIV treatment adherence by postpartum women living with HIV

Katy Pepper[1,2]*

1 Community Engagement Division, Rhodes University, Makhanda, South Africa, 2 Ubunye Foundation Trust, Unit 7, Makhanda, South Africa

* k.pepper@ru.ac.za

## Abstract

### Introduction

Initiatives to support adherence to HIV treatment in South Africa are often centred on service delivery thereby avoiding key challenges to adherence: stigma and poverty. In contrast, this study aims to demonstrate the strength of an inclusive research and programme approach to improving the lives of people living with HIV and simultaneously ARV adherence.

### Methods

Participatory Action Research combined with a visual participatory method (Photovoice) was used by postpartum women to share their experience of taking ARVs. The research was analysed from an interpretative and critical paradigm where both the women and a non-governmental organisation collaborated in the data collection, analysis and interpretation of the findings. Together, they then disseminated the findings and used a community-led approach to create a programme addressing these barriers effectively.

### Findings

Two main barriers to ARV adherence emerged: the anticipated stigma associated with issues of disclosure and poverty epitomized by alcohol abuse, gender-based violence and hunger. The women and the NGO staff successfully presented their findings at conferences and collaborated to develop a programme of support for all women living with HIV in the area. The programme addresses each of the issues raised by the co-researchers and is run via a community-led process where the participants lead on design, implementation, and monitoring and ultimately will revise the programme as needed.

**Data Availability Statement:** The data underlying the results presented in the study are available

from The Ubunye Foundation Trust
(admin@ubunyefoundation.co.za).

**Funding:** The author received no specific funding
for this work.

**Competing interests:** The author has declared no
completing interest exists.

## Discussion

The inclusive approach of this study enabled these postpartum women to portray the intersectional nature of both HIV stigma and poverty that affects their lives. By working with the local NGO to develop a programme based on these insights, they were able to tailor specific interventions to the issues women living with HIV face in their area. In doing so, they aim to improve the lives of people living with HIV by demonstrating a more sustainable way to impact ARV adherence.

## Conclusion

Currently, health service insistence on measuring ARV adherence does not address the core barriers to taking ARVs and misses the opportunity to focus on the long term health and well-being of people living with HIV. In contrast, locally targeted participatory research and programme development based on inclusivity, collaboration and ownership do address the fundamental challenges of people living with HIV. In doing so, it can have a greater impact on their long term well-being.

## Introduction

Health care services' interventions aimed at addressing non-adherence to antiretrovirals (ARVs) by those living with HIV in South Africa are often derived from a single source, health services research. HIV health care services pick their targets of the support they provide for adherence from the findings of this research. Health service research frequently recommends a focus on commodity-supply changes such as ARV supply-chain improvements or using community distribution networks, dispensing machines, and telephone reminders [1, 2]. Some interventions have centred on differentiated care for self-help groups and adherence clubs [3]. *What would happen if people living with HIV led the research process and based on their findings created a programme to address those barriers*?

Often, the impact of HIV treatment and care services in South Africa are measured in terms of ARV uptake and viral suppression and ranked against the UN 90:90:90 goals [4]. Yet HIV viral suppression rates are not as static and upward as we imagine. Reported adherence rates or numbers returned to care are a glimpse of a moment in time, rather than the everyday lived experience of those living with HIV. *Measuring success in these ways might justify the service intervention but what if success in adherence was measured in a different way, one that focuses on the life of the individuals using the service*?

This article examines one type of combined research and intervention model and ascertains its effectiveness in answering these questions. The paper describes a collaborative, person-centred approach to understanding the barriers to ARV adherence that forms the basis of a local programme of support for women living with HIV. To test this model further, the study focuses on the most salient example of people with high rates of non-adherence, who are also the most affected by social and economic constraints—postpartum women. This trend in non-adherence by women postpartum is found across the globe [5] including in South Africa [6], where 50% of women are lost to ongoing HIV care by 6 months postpartum [7].

## Background

Sakyi *et al...* found that the increased tendency of postpartum mothers to be lost to HIV care was due to two factors: the experiences of poverty and HIV stigma [8]. This is confirmed by systematic reviews and meta-analyses of the many studies of pregnant and postpartum women living with HIV [9].

Poverty levels are the highest amongst female-headed households. StatsSA points to a 2.3% increase in the poverty gap to 23.2% for female-headed households in 2015, compared to a less than 1% increase to 13.6% in the poverty gap for male heads of households [10]. Women in South Africa are up to 30% poorer than men, with women making up the highest number of unemployed people [11]. Statistics SA (StatsSA) reported in 2017 that poverty was highest amongst women living in the Eastern Cape [10], whilst the province itself had the highest score in the country on the South African Multidimensional Poverty Index, and twenty of the poorest municipalities [12]. The poverty differential between men and women is related to women's ever-growing rates of unemployment, the increasing concentration of women in low paid and informal jobs, and the decline in the number of women getting married (fewer women with access to male income [13].

Several studies reveal a strong link between poverty and poor adherence to ARVs by women living with HIV [14]. The studies indicate that the effects of poverty, particularly food security, inadequate housing and unaffordable travel costs, meant women struggled to take their ARVs regularly [15]. The link between poverty and food insecurity is undeniable. The most common reason for missing ARV doses is a belief of severe illness if the pills are taken on an empty stomach [16]. When food supplies are extremely scarce, ARV adherence may cease entirely [17]. The poverty experienced by women triggered by their unequal access to education and employment opportunities means they often feel compelled to stay with their families, or with a male partner, even when this may risk violence. A longitudinal study of South African women postpartum, found that those living in poverty were more likely to experience, and were more vulnerable to, intimate partner violence (IPV) and the accompanying psychological distress than those with more financial stability [18]. This in turn increases their risk of failing to take their ARVs at the time of the violence and the ensuing depression [19].

People living with HIV often describe HIV stigma as being the main barrier to them seeking care and remaining on treatment. This stigma can be divided into three types of stigma experience: "internalised" stigma, a person's assimilation of stigmatizing beliefs leading to feelings of shame and anxiety associated with being HIV-positive or on ART; "externalised" stigma based on a person's direct experiences; and "anticipated" stigma based on a person's fears of prejudice and discrimination based on HIV status [20]. This experience of anticipated stigma is well documented as being a major factor in controlling the taking of ARVs by people living with HIV. In their review of research into why people newly diagnosed with HIV do not start their treatment, Ahmed *et al.* show how those experiencing anticipated stigma felt that starting ARVs would broadcast their status, putting them at risk of stigmatised reactions from others. These reactions include social rejection and isolation by family, current partners, peers and the community around them [20]. Peltzer *et al.* in their longitudinal study of stigma experienced by pregnant women living with HIV found that anticipated stigma was associated with alcohol abuse, depression, decreased HIV knowledge and non-disclosure of HIV status [21]. For mothers living with HIV, their fear that HIV would compromise how they are perceived by family and others often delays disclosure of their status to their children or other close family members [22].

Yet postpartum women living with HIV provide one further unique characteristic which, paradoxically, provides an extremely strong counter-pressure in favour of adherence: these

women are fully committed to the need to take ARVs consistently–many have had the key experience of having seen the protective strength of ARVs in ensuring their children are not infected. They know that their route to health and wellness is by maintaining full adherence to their ARVs. In this way, they can ensure the continued care of their children [7, 23].

To catalogue and bring all these experiences and perceptions together but most importantly to use that information to work on relieving the situation for all postpartum women living with HIV requires a different research and programme development approach. Three years ago, writing in the Journal of the International AIDS Society, Pantelic *et al.* argued for a re-orientation of research such that "it places people (rather than intervention or disease) at the centre of our response". They outlined a model of community-based participatory research that involves "planning, executing, and disseminating research *with* the people whose life-world and meaningful actions are under study". They also emphasise ensuring more person-centred research which allows outcomes to be measured by what matters to the individual using the service and not, for example by a reduction in viral load. Most importantly they called for a re-definition of 'participatory' whereby people living with HIV would not merely be participants but rather essential technical advisors, partners in the research (and programme design and implementation), co-owners of the data and key stakeholders for dissemination [24]. This Photovoice study is an attempt to provide that kind of research: it was developed to enable postpartum women to not only unpack the barriers they experience in taking ARVs but to address those barriers by disseminating their experiences to others and developing a programme of interventions designed to tackle them.

## Methods

### Participatory action research using the photovoice method

The study described in this paper follows Participatory Action Research (PAR). PAR contains four essential elements: *Participation* of those who are the focus of the research, in collecting knowledge through a systematic *Research process*, resulting in *Action* that leads to Social Change [25]. In essence PAR re-arranges and blurs, the roles and responsibilities of a traditional research process turning the researched into researchers. In PAR "knowledge is co-created, acted on, and learning from action is sought to bring about and sustain change" [26]. The method used in this study was Photovoice. Photovoice enables people to use the photographs they take to express and share their views. It is a method that has been widely used in health settings and particularly in research with people living with HIV. Teti *et al.* reviewed 22 studies using Photovoice with people living with HIV and indicated that the method can enable them to "inform research and practice priorities. . .but that more studies should be undertaken resulting in action" [27]. In a review of reviews of thirty years of Photovoice use in health research, Seitz and Orsini also identified a lack of following through to action. They also found inconsistencies in the way the method has been used in these studies. A central finding was a lack of participant ownership of a project. The studies did not follow through on ensuring co-researchers collected the data, analysed the data, and published and presented findings. They also pointed to an absence of descriptions of the ethical procedures used and the value placed on co-researcher well-being [28]. In this Photovoice study with postpartum women living with HIV, there has been a concentrated effort to avoid these inconsistencies and to provide the co-researchers with not only a public forum where they can describe the barriers that they face in taking ARVs but also lead to a programme to address those barriers.

### Study context

The study took the research section of the Participatory Action Research was conducted in 2018–2019. In the action section, the programme development and implementation are

ongoing. in Ngqushwa Municipality, part of the Amathole District of the Eastern Cape of South Africa. The population is approximately 4,580 people with 1,330 households. The villages were created in 1994 by the then South African apartheid government's forced migration policy. This saw thousands of people moved from their homes to largely deserted rural areas with no infrastructure, support or communication networks. Even today, the villages are only accessible via poorly maintained gravel roads. It is a very dry and barren area Most of the young people leave to find work in the towns and cities. No children can complete their schooling in the area since there is only one secondary school with no mathematics or science teaching. The vast majority of the 15–64 age group in this area are "not economically active" [29]. In 2009, nearly 18% of all young women aged 13–18 years in the province had been pregnant, with many becoming pregnant before the age of 16 years [30]. In 2012, HIV prevalence rates in the province were 19.9% among 15-49-year-olds. An antenatal survey in 2013 indicated that 31.4% of all pregnant women who visited a clinic in the Eastern Cape were living with HIV [31]. These rates remain high nearly a decade later.

The impetus for this study was a clinic facility manager asking a local NGO to help them address the rising number of women living with HIV, who had recently given birth and were "defaulting" on their ARV treatment. The ten women volunteers who took part in the PAR study and subsequently led the programme were from four clinic catchment areas. The local NGO is the Ubunye Foundation which follows an Asset Based Community Development approach used by a growing number of NGOs in South Africa. This approach emphasises the mobilisation of local community assets and skills [32] and like many NGOs using this approach, the Ubunye Foundation's governance structure is based on *community-led* programmes [33].

This study, although facilitated by members of the Ubunye Foundation, was a partnership between the Ubunye Foundation, and the 10 women volunteers (co-researchers) who used the PAR method Photovoice to document their barriers to ARV adherence, and the community of women living with HIV who set up the programme. This partnership was based on transforming the women being researched into researchers, and the women who were participants in a programme into leaders in the design and implementation of that programme.

## Ethical considerations

Ethical approval for the Photovoice research was obtained from the Biomedical Research Ethics Committee (BREC) at the University of KwaZulu Natal (BFC253/17). This research followed an explicit and detailed process aimed at ensuring both the postpartum women co-researchers and the Ubunye Foundation staff were equal collaborators in conducting this study. The inclusion of the Ubunye Foundation was specifically targeted because of its community-led approach to programme development -design, implementation and monitoring and evaluation. Therefore, the preparation for the study was focused on not only introducing but actively ensuring the co-researchers ownership of the data, outcomes and the programme that followed. In addition to this ethos of ownership, protocols for ensuring the co-researchers welfare were strictly adhered to including sessions on the study methods, co-researcher and Ubunye staff roles and responsibilities all with specific attention to co-researcher's welfare throughout the study process and beyond. Formal consent forms in isi-Xhosa were explained with ample time for queries and clarification before the co-researchers chose whether to sign. Consent centred on the co-researchers' participation in the study as fully informed and voluntary. It also allowed for the recordings of all study discussions and photographs to be used primarily for programme design but also for dissemination to wider audiences at conferences, public meetings and peer-reviewed journals.

Confidentiality in the study was a key concern. As discussed below, the co-researchers changed the Photovoice process in a way that they felt they could participate in the study, their photographs could be seen and discussed by others, but they could keep control over who ultimately was informed about their HIV status. As additional protection of their identity, the photographs taken by the co-researchers and their recorded photograph discussions were anonymised using a code number, year and session number. There was no monetary payment for involvement in the study and programme design and implementation. However, opportunities were given to all team members to present at conferences or workshops and for these trips, they were reimbursed for transport costs, accommodation, meals and incidental expenses.

## Study participants' selection and roles

The study was made up of two different groups of collaborators or members:

The women co-researchers were all isi-Xhosa-speaking and had attended one of the 4 clinics in the Ubunye programme area at least once for antenatal care. The women were selected based on 3 criteria: first, they were living with HIV; second, they had given birth in the previous 3 years; and third, they had been diagnosed and initiated on ARV treatment during their pregnancy. A month was set aside for recruitment and women meeting these criteria (based on records held at the clinic) were informed about the study by the local clinic staff. Of the 37 women who met the criteria, ten women were randomly selected and all of them agreed to take part in the study. One participant left the study after the orientation and training and another participant was randomly chosen to replace her and she agreed to join the study.

The co-researchers' role was to finalise the data collection process, provide the Photovoice data (the photos and the de-brief discussion), and review and analyse the data. They were also joint disseminators of the study findings and, ultimately, advisors on the programme design. Table 1 gives the status of these 10 co-researchers at the start of the study, with the anonymised numbering that was used.

The other group of collaborators from the NGO were selected because of their work on the health programme at the Ubunye Foundation. Only one of these staff members is living in the study area. She is an isi-Xhosa woman, born and still living in the area. She is a trained Community Health Worker and worked as a Field Officer for the Ubunye Foundation and as a study facilitator for the study. The other two study members live in the nearby town of Makhanda. One is a Professional Nurse who worked as a Programme Coordinator for the Foundation and was a Study Facilitator. She is also an isi-Xhosa woman, born in the Eastern

**Table 1. Status of the ten co-researchers.**

| Study No | Residence | Age | Period Postpartum | Role | HIV status of infants |
|---|---|---|---|---|---|
| 44–18 | Glenmore | 26 years | 10 days | Co-researcher | HIV negative |
| 52–18 | Glenmore | 39 years | 11 months | Co-researcher | HIV negative |
| 24–18 | Glenmore | 21 years | 15 months | Co-researcher | HIV negative |
| 54–18 | Ndlambe | 28 years | 12 months | Co-researcher | HIV negative |
| 14–18 | Ndlambe | 43 years | 21 months | Co-researcher | HIV negative |
| 42–18 | Ndwayana | 24 years | 21 months | Co-researcher | HIV negative |
| 12–18 | Ndwayana | 31 years | 24 months | Co-researcher | HIV negative |
| 22–18 | Pikoli | 29 years | 7 months | Co-researcher | HIV negative |
| 34–18 | Pikoli | 45 years | 11 months | Co-researcher | HIV negative |
| 32–18 | Pikoli | 41 years | 21 months | Co-researcher | HIV negative |

Cape. These two study facilitators' roles were to set up the study in the local Ubunye Foundation programme areas, facilitate the Photovoice data collection, coordinate the data cleaning and analysis and share in the writing up of results and dissemination. The third study member, the author, is an English woman, living in South Africa for 15 years. She was employed by the Ubunye Foundation as a Programmes Manager. She was the research lead for the study, managing the overall study process. She ensured the study protocol was completed and that ethical approval was submitted, cleared and adhered to. She used Nvivo for the qualitative analysis, coordinated the sharing of key themes from the data analysis and the feedback, writing up and the dissemination of findings. The Ubunye Foundation facilitated the programme development process that emanated from the Photovoice and generated funding for the programme interventions.

## Study procedures

The entire 'Research to Programme' process was completed in nine stages divided into two parts (see Fig 1).

**The Photovoice research process.**   In preparation for the research process with the co-researchers, the lead researcher conducted a three-part training session with the Ubunye research associates. These sessions were conducted over several weeks before the start of the study and were usually 1–2 hours. The sessions covered three areas: aims, approach and practical considerations for the study; practical use of the cameras, taking photos, introducing and encouraging the Photovoice technique with co-researchers, and arranging the 4-week study cycle; training and mentoring the staff in running the weekly debrief sessions with the co-researchers.

The next step was for each of the 10 women who agreed to take part in the study to participate in an orientation and training about the Photovoice study aims and processes. The training covered three sections: the use of the camera; rules for taking photographs; and orientation to their work as co-researchers. Written guides in isi-Xhosa were shared for each section, and participants were encouraged to ask questions and practise with the cameras. The session on camera use was on how the camera worked and how to ensure the photos they took were stored. It also included ensuring their safety as they used the camera and discussions on how to explain their photo-taking to others without risking their HIV status. The main rule for taking photographs was ensuring all the photographs taken did not include faces or any image that could identify another person. This included their image and that of their family and friends. The session on their work as co-researchers centred on thinking through how they would take photos on the research theme "the motivators and barriers that they experience or have experienced when taking their ARVs". They were encouraged to think about events in the past and current times when they have been taking their ARVs and the scenarios which make it easy or difficult physically or emotionally to take their ARVs.

For the study period, all the co-researchers were given cameras to use for over 4 weeks. At the end of each week, they met with the Ubunye staff member and went through the photos they had taken. These debrief sessions were recorded. The co-researcher was able, at any time, to remove a photo from the collection or stop the recording or not discuss a photo. The aim was for them to choose what best illustrated the issue they wanted to discuss about taking their ARVs.

**Changes to the Photovoice process.**   The Photovoice stages described in Fig 1 are based on Wang and Burris' methods, and outlined by Leibenberg [34] with one exception related to the Photovoice goal: "To promote critical dialogue and knowledge. . .through *group discussion of the photographs*". Concerns of the co-researchers around confidentiality and maintaining

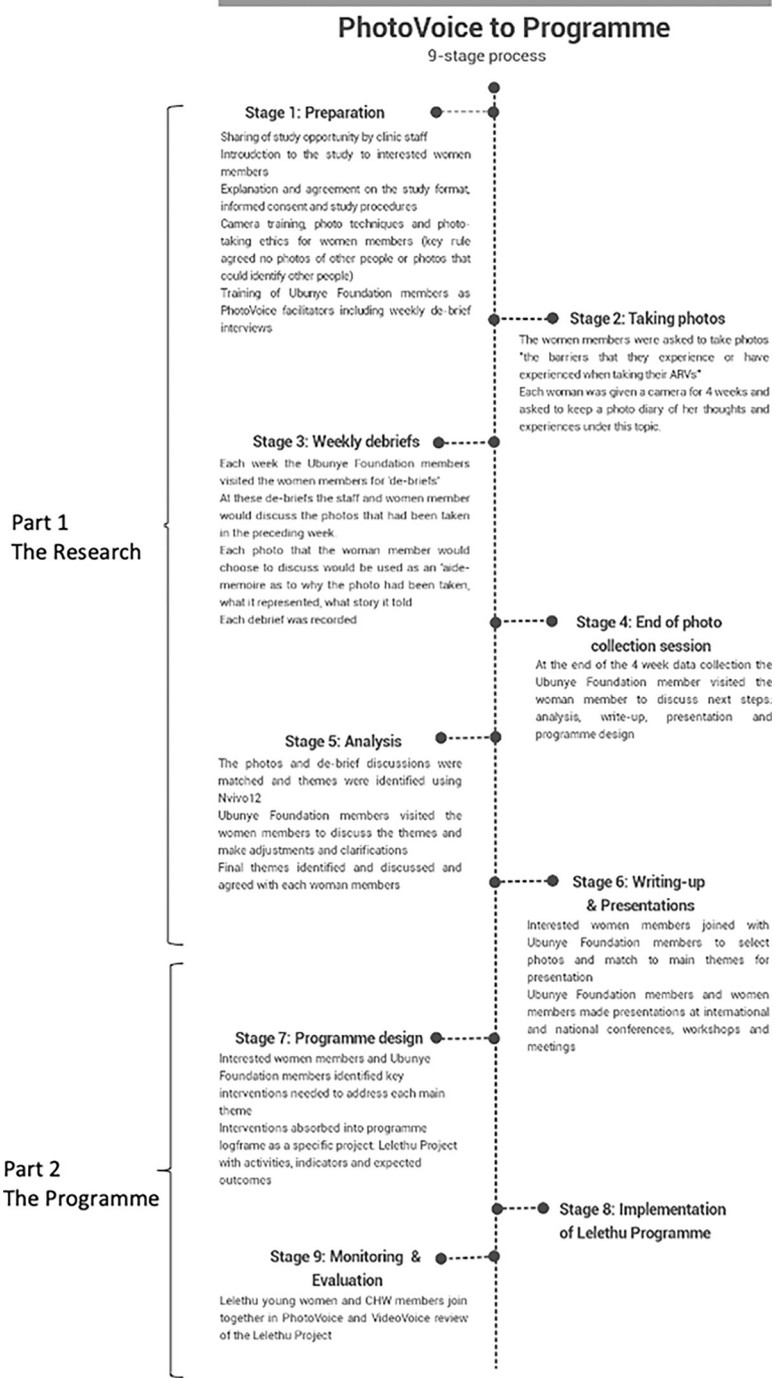

**Fig 1. Photovoice to programme stages.**

anonymity, even between the 10 co-researchers, led to an alternative process of discussion: photograph discussion sessions were completed individually between the women and the Ubunye Foundation members each week. Another adaptation was made at the end of the study. Rather than group discussions each co-researcher reviewed the entire research

photograph and photograph discussion data with the Ubunye members individually. At these individual sessions, they also clarified their own and examined the other study members' inputs.

**Data analysis.** After each debrief session the photos on the camera and the recording were encoded and saved on a hard drive. The recordings were transcribed and together with the recordings were given to external translators (Masters Language students) to translate into English. The transcriptions were reviewed by both the Ubunye staff team and the co-researchers for accuracy. This was done at further individual co-researcher visits. During these visits, the co-researchers also reviewed the work of other co-researchers and the similarities, trends and differences they identified were recorded. The photos and English transcriptions were then coded using Nvivo, a qualitative research application which traces and identifies trends in the data. Once the trends and main themes had been identified the Ubunye staff and co-researchers met again in individual meetings to review them, discuss their accuracy, and implications and agree on the final findings.

**Participatory Action Research–Action process.** *Dissemination of results and advocacy and programme development.* The Action part of Participatory Action Research is an essential part of the methodology. For this PAR study, Action centred on two activities: first, dissemination of findings and advocacy. Once the findings had been agreed upon by the co-researchers and Ubunye staff, abstracts were jointly developed and submitted to international and national conferences, meetings and workshops. The results section documents the success of this process and the joint presentations developed and presented by the collaborators.

Second, the co-researchers and the Photovoice data designed and played a leading role in the implementation of a programme to address barriers to adherence among women living with HIV in their area. This Action was structured around the Ubunye Foundation's community-led governance system, Siyakhana. Siyakhana ensures that programmes are designed and led by those who are at the core of the programme's purpose. At the grass-root level are Programme Working Groups (PWGs). PWGs are made up of up to seven community representatives (who consult with the wider community) and two Ubunye staff members. Each PWG is responsible for designing, implementing, monitoring, adapting and revising the programme as required. The co-researchers contributed directly to the work of the Programme Working Group (PWG) and led the programme design via 3 steps:

Step 1: Co-researchers agreed on the key findings of the study and identified the areas for interventions. This was done during the Photovoice data analysis discussion process outlined above.

Step 2: The PWG (which included co-researchers) met to review the intervention areas and design the specific programme activities, targets and expected outcomes.

Step 3: Ubunye sought funding for the programme and with the PWG identified the local resources that were to be mobilised for the programme.

The outcomes of this process are reported in the results section below.

## Results

### Participatory Action Research: Research findings

**Barriers to ARV adherence.** The co-researchers identified two main barriers to taking ARVs: Anticipated stigma and poverty. At times these two barriers were insurmountable. It is important to note that none of the barriers cited relates to physical barriers such as lack of access to supplies, including clinic stock-outs, or an inability to get to the clinic. The section also reports on the findings of the Action part of the Participatory Action Research that of the dissemination of results for advocacy and programme development.

*1. Anticipated stigma.* The co-researchers in this study spoke of stigma as being the single most powerful determinant of the way they manage their lives. The stigma they described was predominantly "anticipated' stigma" defined as being based on a person's fears of prejudice and discrimination because of the HIV status [20]. The co-researchers spoke of having to carry the burden of this disease secretly and silently. This was their method of protection, their defence from the potentially devastating effect of others knowing their status. For one woman co-researcher, HIV was her badge of shame, out there for all to see, and although she did everything to hide her HIV status, she felt that HIV was somehow so all-pervasive that it would be obvious to others that she was living with it (see Fig 2):

*I had a long trip to go to this hospital. I had a problem [taking my medication from there] at first because I used to feel like it's written on me, that this is what I am there for.*

*Yes, I had to take a taxi there because I was ashamed of having the virus. . ..I felt I didn't take care of myself and I was ashamed.*

The co-researcher goes on to explain that it was the anticipated reaction of others that was the key motivator for the hiding and secrecy. The thought of others finding out was unbearable:

*Eish, I wonder what my husband is going to say when he hears that I'm like this or I wonder if my sister, will she love me again when she hears that I'm like this? Then again I say, I wonder here at home aren't they going to hate me?. . .*

*For an example when I'm doing the dishes or washing a spoon I have used, aren't they going to say they don't want to use the same spoon, because when this disease started people used to have to use their own plates?*

*(34–18 4$^{th}$)*

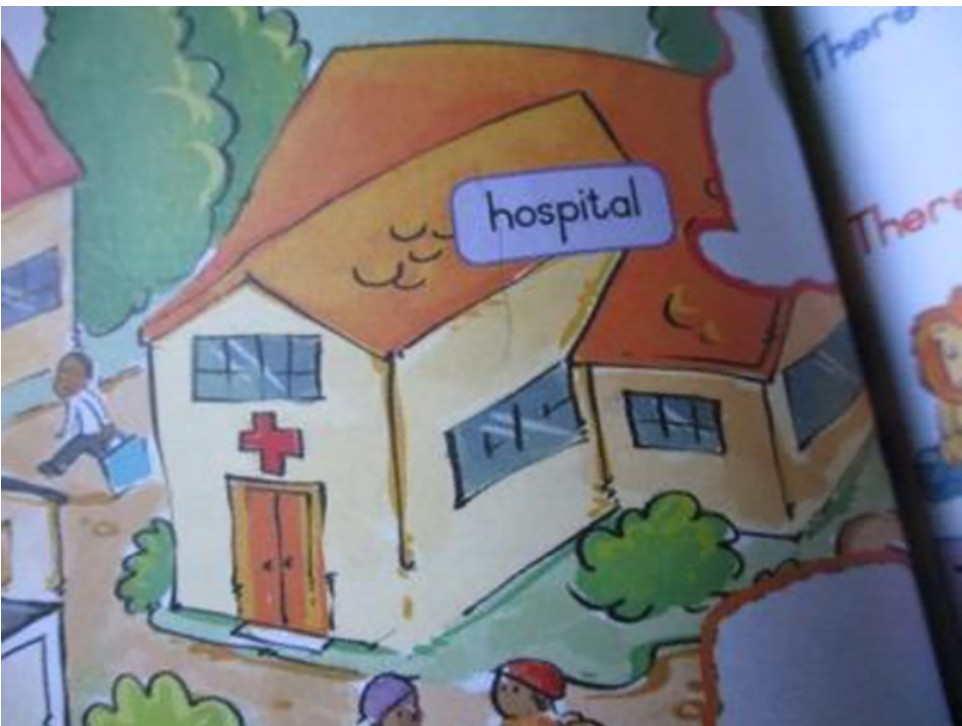

**Fig 2. Image from co researcher 34–18.**

This fear of family, friends' and partners' potential reaction to their HIV status rules the co-researchers' lives from the moment they know their diagnosis. ARVs are seen as the "big shiny beacon" that will tell everyone [that they were living with HIV]. This fear alone was the most frequently mentioned reason for not taking ARVs. Since the medication is taken every day, the ARVs themselves represent a constant threat of exposure. The co-researchers spoke of being preoccupied each day with collecting, storing, taking and managing their ARVs (see Fig 3):

*I have photographed this bag because when I go to the clinic I carry it. But what worries me is that people all the time want to know what makes me go to the clinic with a bag as huge as this one. And I say to them, because now I have a baby I put the napkins and the food stuff for the baby in it. And they respond by saying 'Get away, she is lying, she is lying,*

This fear extends to their home where no one knows their HIV status (see Fig 4):

*Here is a house maybe. When you are at home and you afraid like maybe [you are the] only one person in the family that knows about what you live with.*

*So when there are other people you are afraid to take your treatment in front of them. Also you must find a private place to put them so that no one sees them.*

*Because written on them are who they are for and some people know what are they for. You are afraid what will they say if they find out.*

The fear also impacts their interaction with others and everyday tasks such as going out shopping or socialising with friends (see Fig 5):

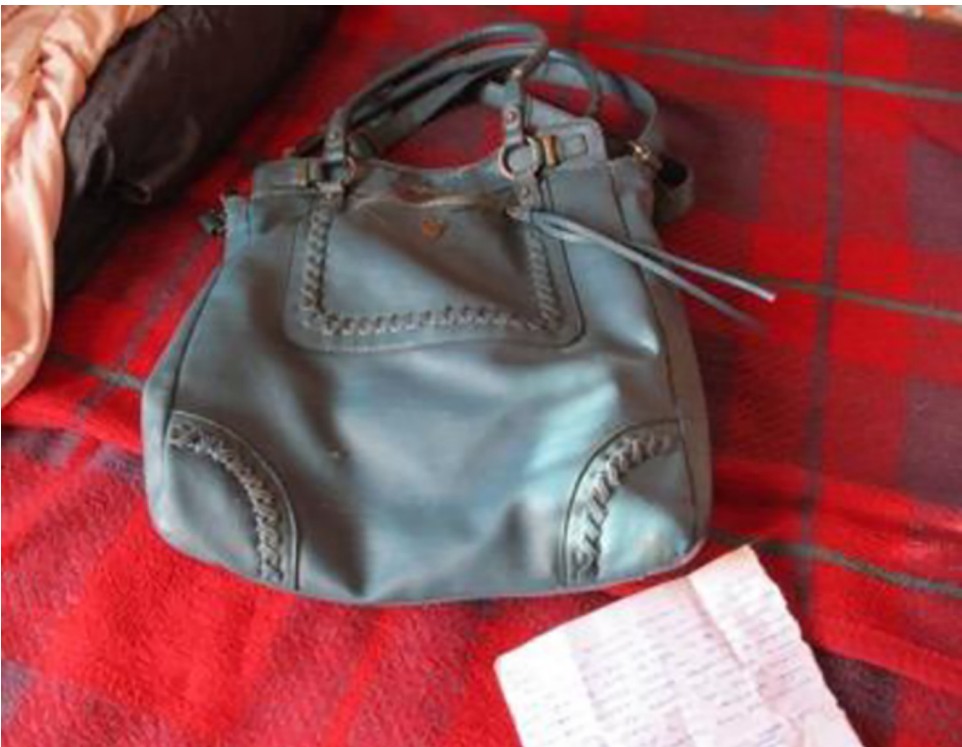

**Fig 3. Image from co researcher 22–18.**

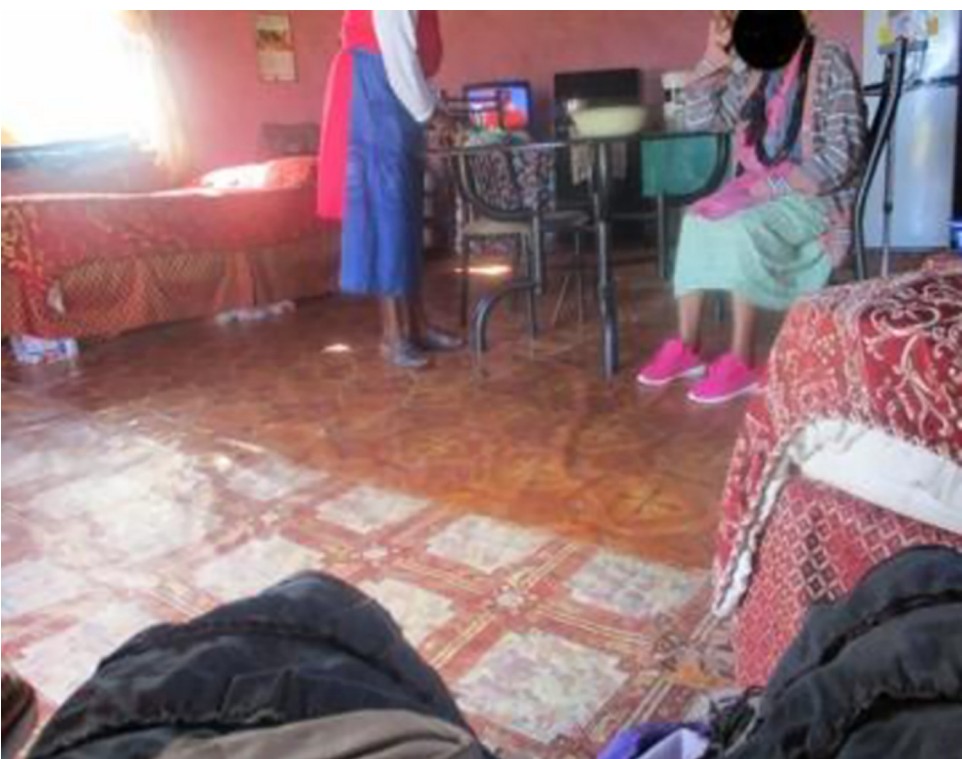

**Fig 4. Image from co researcher 42–18.**

*If I'm with people sometimes I act as if I just need some water and I secretly take it. Not to take it in front of people, I don't have the guts. . .*

*Here it was at night. Someone asked us to help with their car. I couldn't take it [ARV] because I thought what will other people say that I am taking.*

*I had it with me. But I was scared to take it because I'm used [to] taking it at home when I am alone. But I'm scared to take them in front of people.*

Where, when and how ARVs are collected are all critically considered as the co-researchers balance the quality of their lives against the need to take the ARVs (see Figs 6 and 7):

*It is that thing of. . .that. . .that thing that creates worries. They can tell, even without you telling them (HIV status) because of the sound that the pills make inside the container. They make a special sound inside that bottle.*

*(22–18 4ᵗʰ)*

*I hate the noise that is made by the pills it makes me wrong*

*It's my pills I used to be stressed because the sister used to just hold them up and not hide them from everyone else. You see the dispensary is in front of people, so she would pass with our pills just carelessly. I used to be stressed to go to the clinic but in the end they understood me and put my pills in a plastic bag instead of the container. It still worries me though*

A visit to the clinic is often traumatic for these co-researchers since it broadcasts their illness to others attending the clinic and risks their status being discovered:

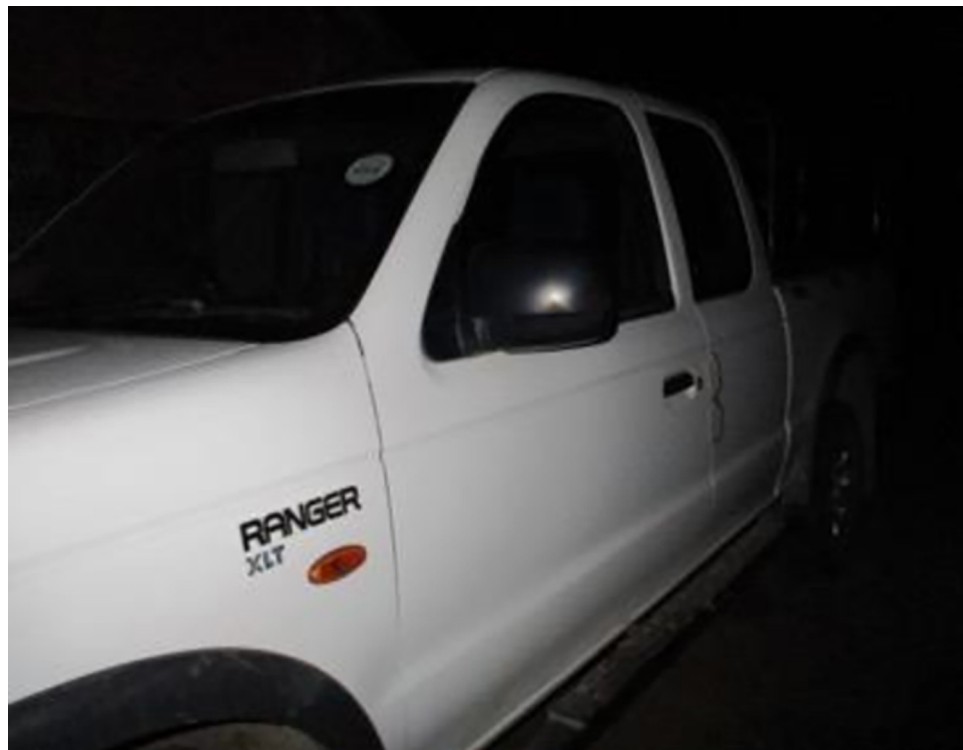

**Fig 5. Image from co researcher 42–18.**

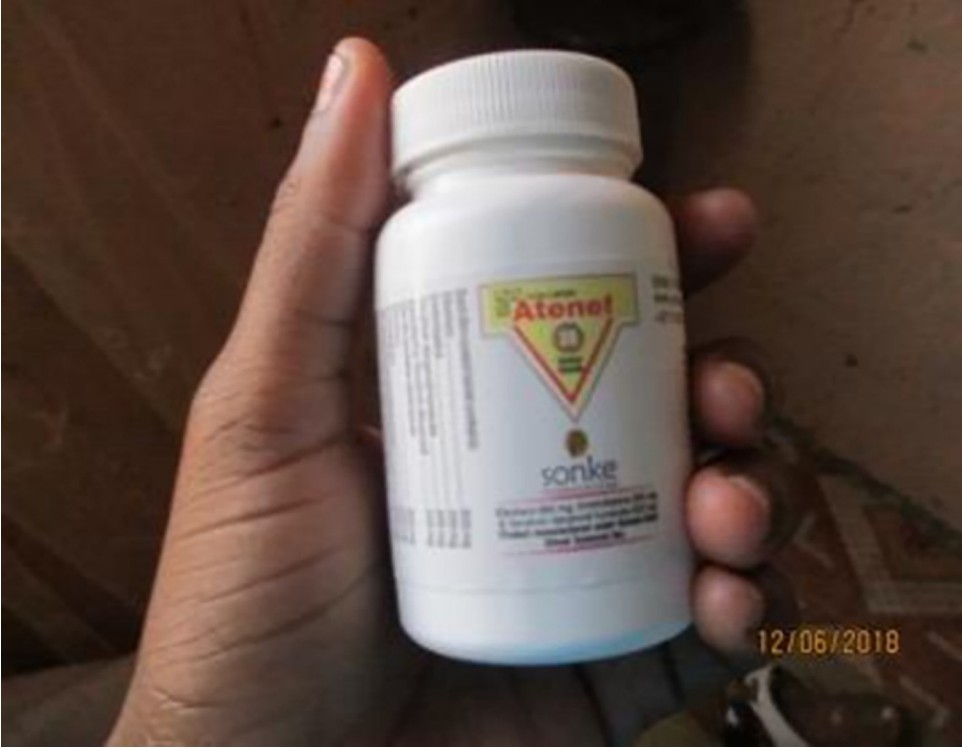

**Fig 6. Image from co researcher 34–18.**

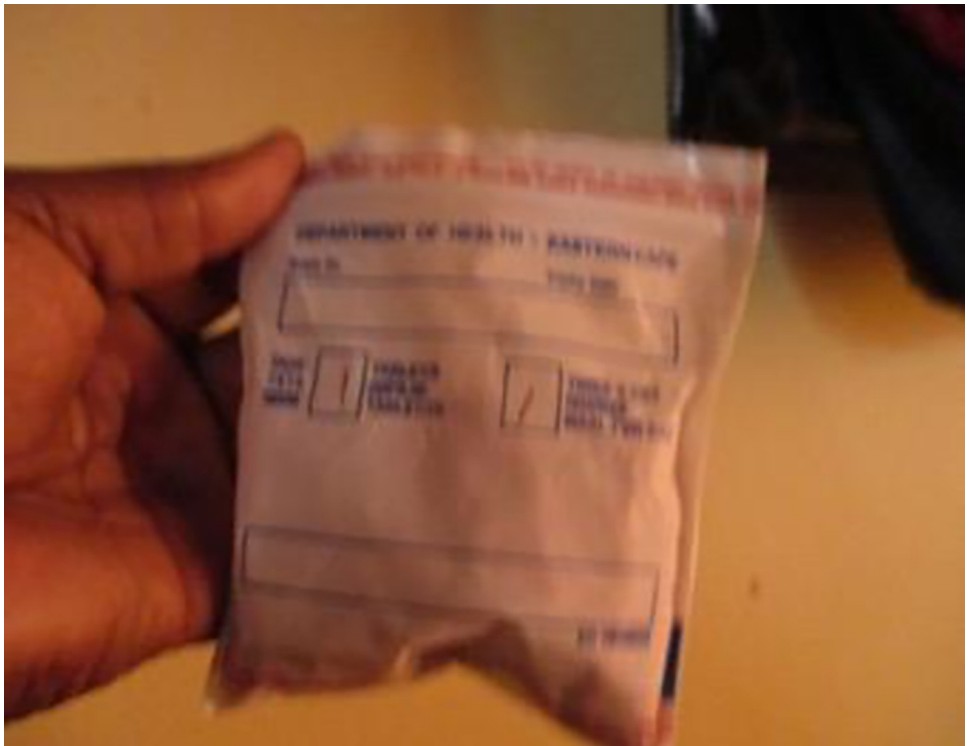

**Fig 7. Image from co researcher 34–18.**

> *So when I was. . .when I found out what that room (clinic room for HIV patients) was standing for, I got [scared] and I started to [be reluctant] to go for my HIV.*
>
> *When someone is taking the pills, some people end up not taking them even if he or she wanted to, because it is said that nurses go out and reveal the statuses of those who were tested [meaning those taking ARVs]. . .I was scared to go take the pills from there because of what people were saying about the nurses.*
>
> *(34–18 4<sup>th</sup>)*

**The HIV stigma "hot spots" surrounding ARV consumption.** The co-researchers described a series of HIV stigma 'hot spots' or situations where they felt at high risk of other people discovering that they were HIV-positive. These hot spots surround the entire ARV pill-taking process: getting re-supplies of ARVs from the clinics; storage of the ARVs at home and even whilst transporting the ARVs from the clinic or outside the home); the packaging of the ARVs which indicate their use; and the space and time for taking the ARVs at home and outside the home. The co-researchers described the lengths they went to avoid others finding out about their status and how these hot spots represented areas where they were at a high risk of their status being revealed. At these times the only way to manage that high risk was to not take their ARVs.

The co-researchers mentioned that clinics had instituted interim solutions to address these hot spots such as fast pill pick-up points, alternative venues for pill distribution, and attention to packaging and labelling. Yet these solutions did not remove the risk the co-researchers felt that their status would be revealed; their anticipated stigma remained.

*2. Poverty.* As in many rural communities in South Africa, there is extreme poverty in Ngqushwa. With few employment opportunities, a challenging agricultural environment and limited infrastructure, communities here struggle to meet basic needs such as food and housing. The women spoke of the added health concerns around HIV making it a very difficult environment in which to live. Maintaining adherence to ARVs is often made more complicated by the daily struggle to find adequate accommodation and sufficient food and clothing for themselves and their children. None of the 10 women had finished their schooling past Grade 10. Each of them struggled to be independent and none of them was able to live away from parents or even an abusive partner or spouse. Without independent financial resources, the women struggled to cover basic needs such as food, shelter, clothes and safety (see Figs 8 and 9).

*Because I'm unemployed, this was a little bit of a grocery. Because these pills are taken after the food, you must be full and I'm someone who is always without food. I look at my pills then just cover them with something and not take them.*

*Here in this photo I'm trying to make this point. The only thing that makes me panic a lot and makes me despair, is the fact of not having food. . .the time to take the pills comes, but then my stomach is empty. I just cover them with something. I know what I'm doing is wrong but it is just, my stomach is empty.*

*The bed in my mother [mother's house]. On the ground [in the kitchen-sitting room is me and my baby, the wind always comes in and the baby always wakes up with a runny nose. There the wind sings inside the house. We are many sleeping in this room. I can't take my pills with all these people around me.*

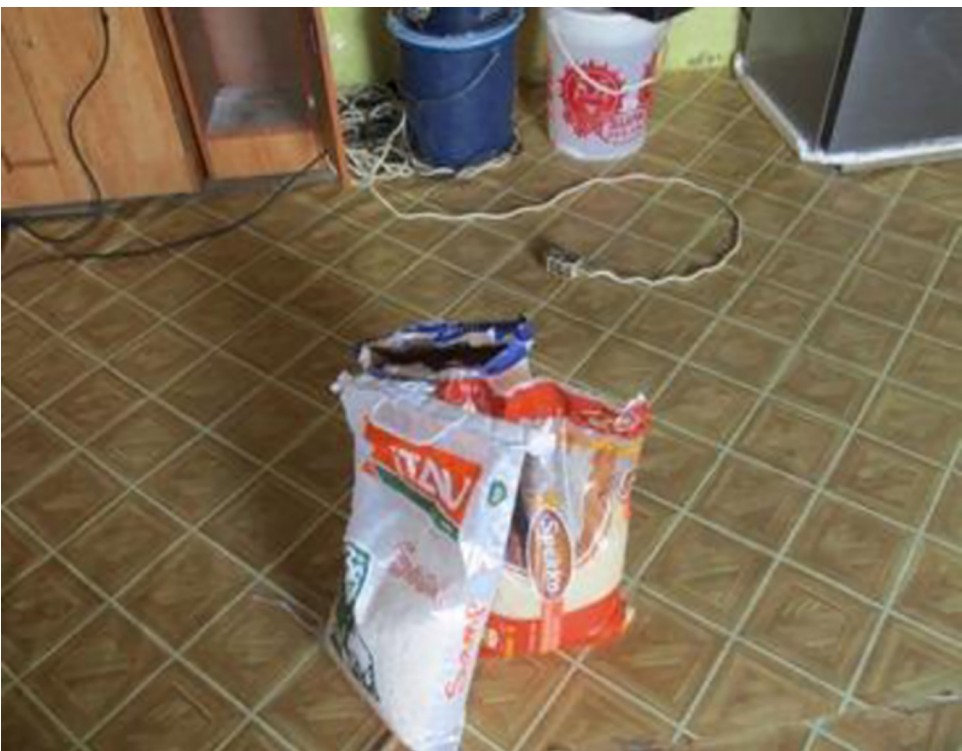

**Fig 8. Image from co researcher 32–18.**

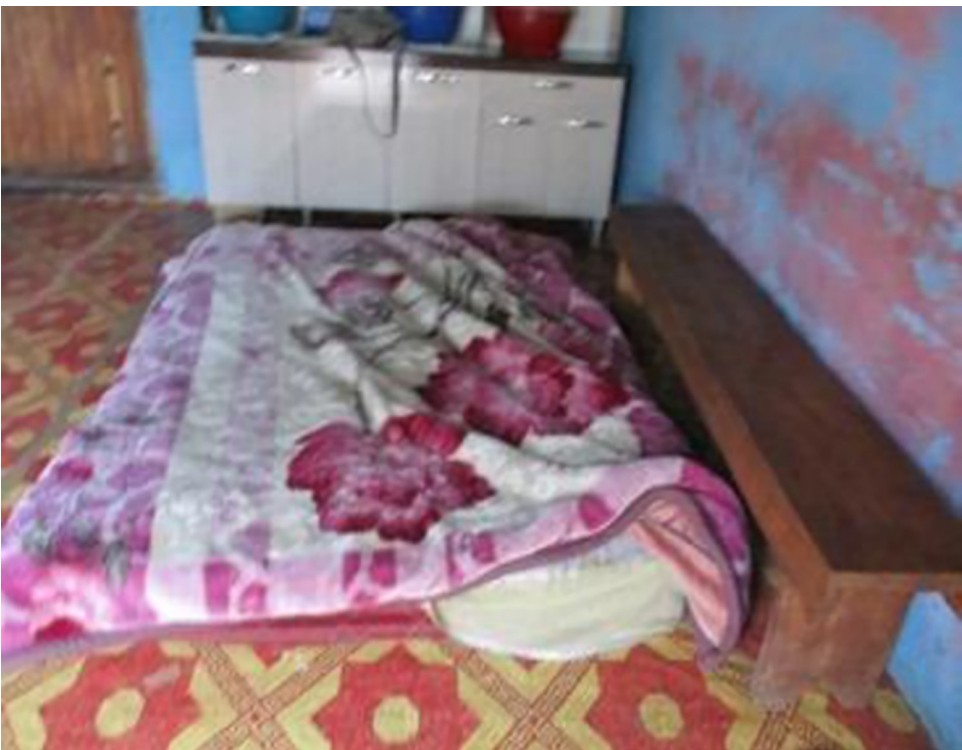

**Fig 9. Image from co researcher 32–18.**

The co-researchers spoke of their dependency on government-provided social grants which were often the only funds that they had to look after their children. This focus on being solely responsible for their children was a constant theme and one that compelled them towards taking their ARVs:

> [Without having a grant, we would have no money] it affects us because when you think about your children you wish they could get things that they need. Sometimes I wish I couldn't see their suffering and die but that won't help as well. Even last week I got a call from my son in prison asking for a warm white shirt–I can't get it for him and I just wished I could die rather than suffer like this. I often would just not take my treatment and wait to die. But then I thought of the little ones and what would become of them.

> (14–18 3rd)

None of the co-researchers expressed dreams of a future, of finishing their education or finding a job. It all seemed too impossible to them, despite desperately wanting change in their lives and craving their independence (see Figs 10 and 11).

> This one is about this house, I want to see my children grow and find work to help us renovate this house or maybe my brother will get a job and help renovate it. Sometimes when we are getting off the taxi people are looking at us. I don't see the need to take pills to live a life like this.

> This is a house this is just a house. I would like to have one. I like the idea of owning my own house. My own house where I will take my treatment without hiding it now and then. I would

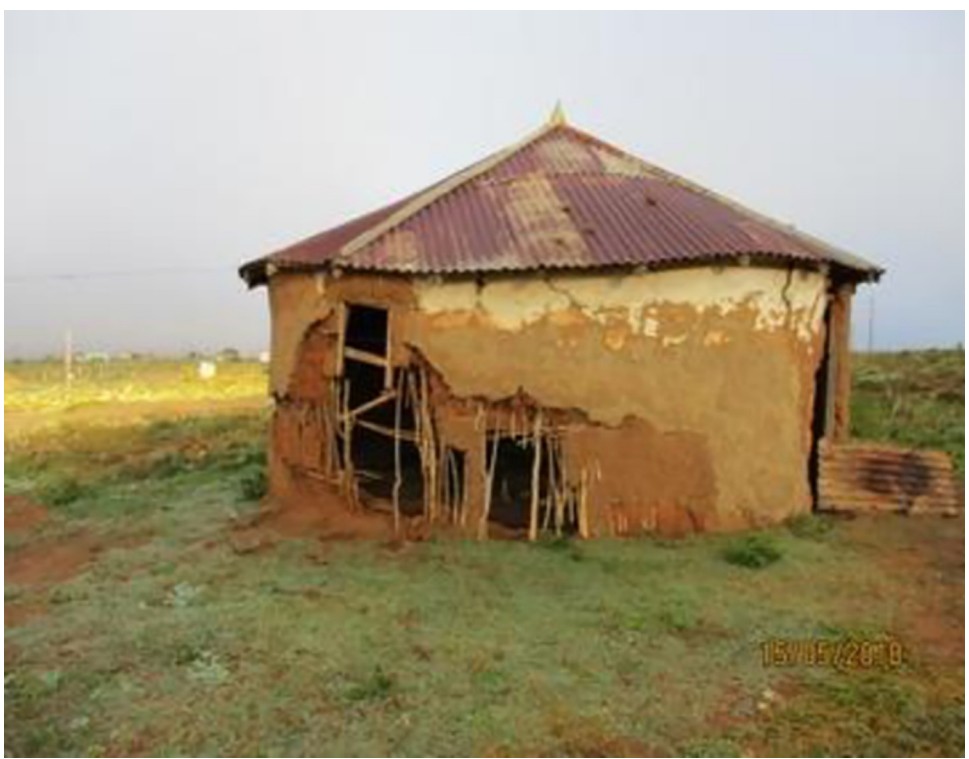

**Fig 10. Image from co researcher 14–18.**

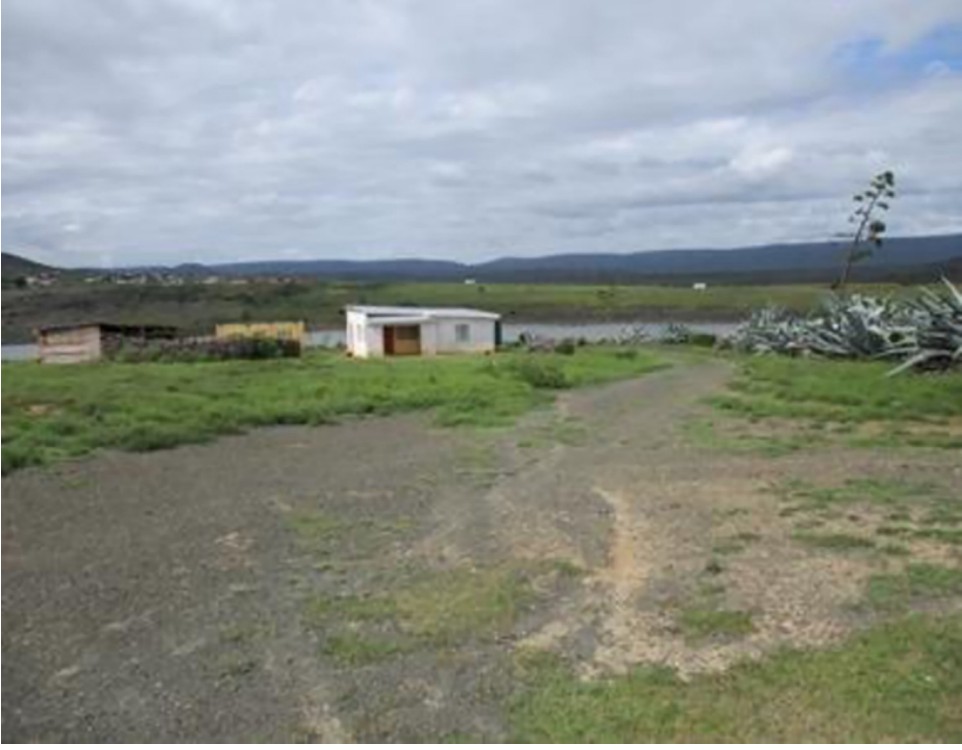

**Fig 11. Image from co researcher 32–18.**

*put it in that wardrobe, in the house where it would just be me together with my child–That is the thing I would like to have in my life to take my treatment very well.*

**Key differences in the perceptions of living with HIV amongst postpartum women.**
The barriers to ARV adherence of anticipated stigma and poverty could be related to many other people living with HIV. However, there were specific differences in the way postpartum women experience and perceive their lives with HIV and these differences ultimately influence their ARV adherence.

The first difference was that the co-researchers had recently given birth and had an infant who was wholly dependent on them for survival. This makes them distinct from others living with HIV. As discussed above, all the co-researchers indicated their strong motivation to stay alive and well for their children (see Fig 12).

This is my child I took this picture because sometimes I ask myself if I don't take this treatment who will look after my child

So even if I feel like not taking my pills I feel I'm forced to take it for my child's sake.

It makes me feel sad because if I don't take my treatment I will die and who will take care of my child. She will be called names and be told "your mother was killed by the virus" things like that

The second difference was that the co-researchers had a strong belief in the crucial role of ARVs which was based on their experience of ARVs keeping their children HIV-negative (see Fig 13).

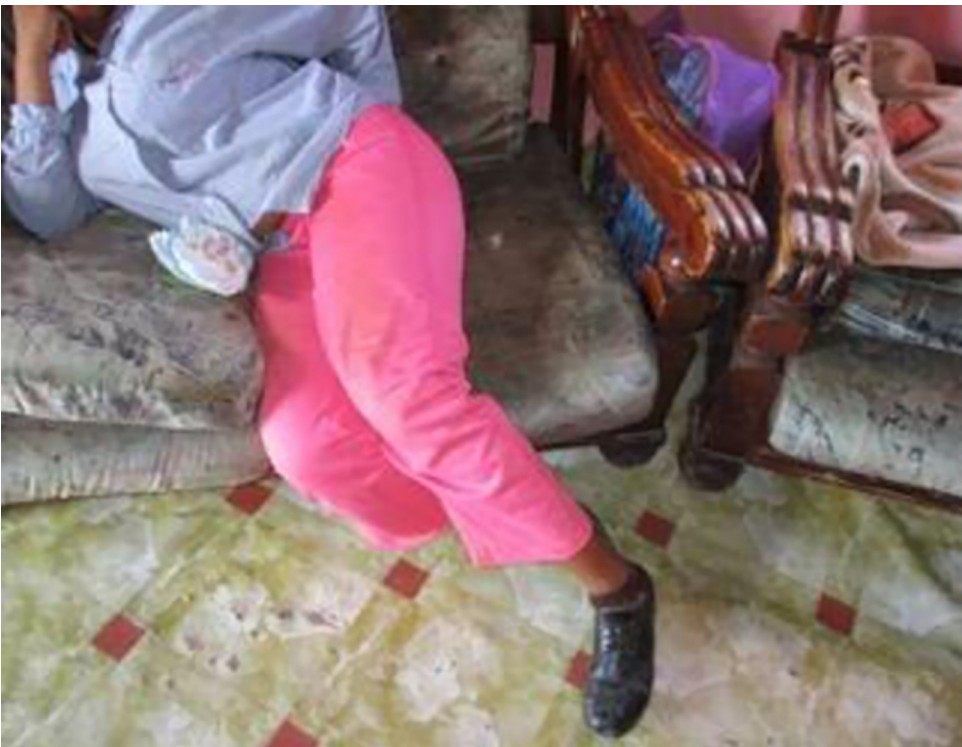

**Fig 12. Image from co researcher 42–18.**

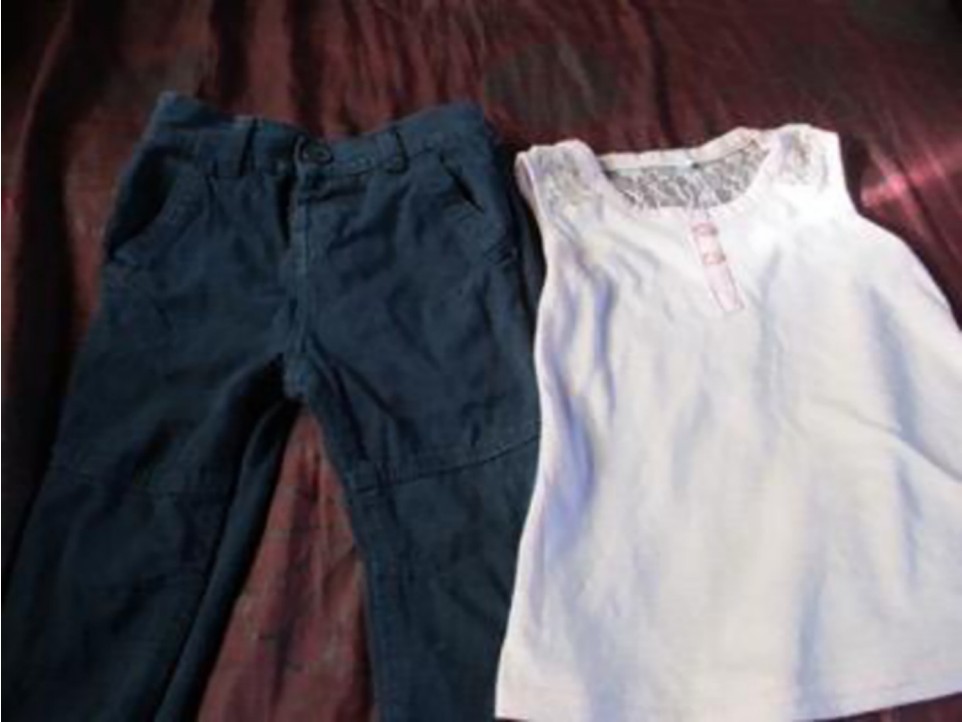

**Fig 13. Image from co researcher 24–18.**

*This photo I have taken it to show that my children did not get infected because of the fact of taking the treatment. It was me that made it bad for them. . .I thought it was my fault*

The third difference was that these postpartum women revealed feelings of extreme guilt at having put their children at risk of the disease. If their HIV status was revealed to others they would not only risk rejection as a person living with HIV but they could also be derided for being unworthy mothers. These postpartum women had risked the lives of their children. Their children would also risk being falsely accused of having HIV, but certainly of having a mother who had the disease. Therefore, the intensity with which these co-researchers guard their HIV status is far more extreme than others living with HIV. It explains their compulsion to remain extremely isolated geographically and socially and also in not having a job. By not seeking employment, the co-researchers increase their dependence on their spouse or family for food, housing and other basic needs for survival with their infants. For these three reasons, these co-researchers fear not taking their ARVs, and every day, they balance that fear against the fear of taking them and being discovered as having HIV.

## Participatory Action Research: Action findings

**Dissemination of findings and advocacy.** In line with the Participatory Action Research methodology, the Photovoice research led to Action. The Action was in two phases, first the dissemination of findings; and second the development of a programme. The dissemination of findings aimed to advocate with policy-makers, decision-makers and programme managers for inclusive, participatory programmes for people living with HIV. It was also designed to inform people both locally and nationally in South Africa about the challenges of HIV stigma. The Ubunye Foundation and co-researchers identified three conferences and a community

research competition to which they submitted abstracts. An abstract was accepted and presented as a poster presentation at the International AIDS Conference in Amsterdam in June 2018 by Ubunye staff and at the Durban AIDS conference in 2019 by both co-researchers and staff. Co-researchers and staff members also presented at the Public Health Association of South Africa conference in 2018. The research team also jointly developed a submission to the Community Chest Impumelelo Awards for which they won a Social Innovation Award in 2018. To advocate for a change in the programme approach locally, the co-researchers and Ubunye staff presented findings at Department of Health HIV programme meetings both locally and in the District. Unfortunately, due to various health programme delays, staff challenges and the movement of co-researchers out of the Province, a jointly written paper for a peer-reviewed journal focused on promoting the programme approach has not been possible. In addition, the co-researchers did not want their names published with the article. Therefore, this article is offered as a collaborative effort of thirteen people who were all key contributors to the Photovoice research, dissemination and programme development.

**Development of a community-led programme.** The other Action emanating from the Photovoice research was the co-researchers leadership in designing and implementing a programme to support other women living with HIV and address barriers to ARV adherence in their local area. As described above this was achieved via the co-researchers leadership in the Programme Working Group through Siyakhana, Ubunye's community-led governance structure. In the 1st and 2nd steps of the process co-researchers with Ubunye staff identified, designed and implemented a set of programme activities designed around the findings of the Photovoice research. Calling the programme, Lelethu (isi-Xhosa for "it's ours") five interrelated programme interventions are being implemented:

1. To address individual experiences of stigma and social and psychological barriers which constrain the adherence to ARVs of people living with HIV, Community Health Workers (government employees attached to clinics) and Ubunye staff worked with women living with HIV to create and follow Life Plans. These Life Plans provide a roadmap to addressing specific difficulties, particularly around family relations, disclosing their HIV status, and accessing clinic care. These Life Plans also detail the steps to women's economic empowerment in [3].

2. To address the anticipated stigma that prevents women living with HIV from accessing health services or taking their pills in public, the Ubunye Foundation supported women who were openly living with HIV to discuss HIV stigma in local schools, health facilities and community organisations.

3. To address the need for economic empowerment for women living with HIV the Ubunye Foundation provides Life Skills training and mentoring which includes support for business development, job searches, applications, seeking and applying for further training and education related to obtaining their ideal jobs.

4. To provide peer support to women living with HIV whilst respecting the need for confidentiality and concerns around HIV status disclosure, WhatsApp groups are run. Women living with HIV receive data [and in some cases mobile phones) to enable them to talk with others, share their thoughts, obtain accurate information on HIV, treatment and care, join together on advocacy initiatives and provide details of jobs, training and education opportunities available.

The effectiveness of these interventions will be evaluated by the Programme Working Group over the coming months using the PAR methodology. However, there have already

been several notable outcomes from the Lelethu programme: of the 118 women aged 16–24 years in the programme 31 women have been supported to find jobs, 11 women started a business and 14 have gone on to further study. Nine Lelethu women have started work advocating against HIV and TB stigma in their local communities; one Lelethu woman was highly commended on her presentation about the Lelethu programme at a multidisciplinary seminar on Gender-Based Violence organised by the French Embassy in Cape Town in 2022.

## Discussion

This study set out with the aim of providing an alternative way to consider ARV adherence from a biomedical emphasis on improving access to ARVs and depending on markers such as pill counts or viral load. Thus the first question in this study sought to determine whether successful participatory research and programme development are more effective ways of tackling the well-being of postpartum women living with HIV. The second question was to ascertain whether improvements in the quality of life of people living with HIV could be a better impact to focus on than ARV adherence. To address these questions the paper has presented an account of a Participatory Action Research study with postpartum women living with HIV in the Eastern Cape of South Africa. The PAR described enabled postpartum women to become co-researchers collecting data on their adherence to ARVs, analysing that data, sharing their findings to advocate for change and working together with NGO staff to develop a community-led programme addressing core barriers to ARV adherence. This study example elucidated 2 answers: first, using a Photovoice research method, followed by Action through dissemination and programme development has produced a powerful, effective and carefully tailored response to the barriers to ARV adherence. Second, using this PAR methodology has provided an alternative and more effective way to view ARV non-adherence. Focusing programme interventions on the barriers to adherence moves measurement of non-adherence away from medical measurement (viral load and pill tracking) to improvements in the lives of people living with HIV. Yet we need to unpick the findings that led to these answers to give a clearer picture of these study outcomes. The theory of intersectionality lends itself to an analysis of the predicament these women have described. Intersectionality maintains that a person's experience is shaped by multiple identities that interconnect and interact [35]. These identities are shaped through social conditioning and reinforced by institutions such as health care services. Campbell and Gibbs talk of women living with HIV as having multiple forms of stigmatised identities as vectors of HIV/AIDS, often poor and socially excluded [36]. The sentiments expressed in the Photovoice data describe women as being at the intersection of HIV-related stigma and gender inequality with a specific focus on women's poverty.

The strength of the HIV-related stigma can be seen in the co-researchers self-imposed isolation from others, their perception (or avoidance) of being seen as a "bad mother", their persistence in remaining hidden and their feelings of having no future. The evidence revealed in the Photovoice accounts describes the co-researchers' intense dread of being discovered as having HIV. Given the severe isolative measures they take, it appears far stronger than that experienced by others living with HIV. The women describe the dread of being exposed as being coupled with feelings of shame and guilt. As mothers living with HIV, they have risked the lives of their infants by being infected with a disease that they could pass on to them. Moreover, despite doing all they could to make sure their infants were safe, they were the ones that put them in danger. This identity of being a 'bad mother' is powerful. It is suggestive of violent acts against children that are viciously condemned by society. So, these postpartum women have carefully concealed their HIV status. However, the daily ritual of taking ARVs, so closely associated with HIV, threatens to expose the women's status. Therein lies the co-researchers' unique dilemma:

these mothers desperately want to take their ARVs since they believe that ARVs offer the best opportunity for their health and longevity so that they can continue to care for their children. Yet taking ARVs risks this exposure to HIV stigma. The co-researchers' response to this dilemma has been to remain hidden and isolated, dampening the risks of both scenarios.

Intersecting the experiences or fear of HIV stigma of these postpartum women is gender inequality. This Inequality is intertwined through all spheres—from the co-researchers unequal access to education, preventing access to jobs, stopping representation in local government and civil society groups and resulting in limited financial resources and poverty. Maintaining adherence to ARVs is made more complicated by the daily struggle to live through this poverty. The co-researchers' vivid portrayal of their experiences of and responses to poverty was striking. Sharing already limited accommodation, having no option to improve their living conditions, limited access to food, unable to buy clothes for their children all these descriptions show the immense challenges these women face. To manage their situation, they are forced into dependency on others and are exposed to the potential abuse that this dependency creates. There is no doubt that these outcomes of women's inequality are exacerbated by living with HIV. HIV stigma plays into their dependency on others with women feeling they have nowhere else to go so they have to keep their HIV status secret in the household to retain their place. The co-researchers' overarching perceived need to remain isolated and hidden seeps through even in their desire to improve their economic situation–wishing to have their own house to be better able to hide their HIV status. The women also talk of a more depressing dilemma, of not wanting to live this life of poverty but having to be around for their children.

Looking at these disparities through an intersectional lens rather than as individual and independent issues enables a view of the overlapping systems of power that affect postpartum women in this unique way. This makes it important that the interventions to address these disparities are created by the women who experience them. They alone can get to the core of how to address the intersecting issues of HIV stigma and poverty amongst people living with HIV in their communities. The women in this study have taken the opportunity to gain greater insight into common experiences they all share. They have been able to give voice to the challenges in their lives and, through an NGO's support, guide the creation of a programme to address them. It is the co-researchers ownership of the process–of participating as an equal partner in the development of knowledge leading to a programme–that is most likely to bring about sustainable change in the lives of women living with HIV in the local area. The almost unmentioned bonus of this process has been the consistent increase in ARV adherence which can be measured not by pill counts, viral loads, and clinic attendance records, but by the improvement in the women's quality of life.

This study has shown that the inclusivity, collaboration and ownership in both research and programme development demanded by Pantelic have enabled women living with HIV to explore their situation and create a tailor-made programme to address the challenges. The impact of the Lelethu programme must be measured using the same Participatory Action Research methods that give ownership to the members of the programme and enable them to lead in its ongoing design and implementation.

## Conclusion

The study set out to investigate first, whether participatory research and programme development were effective ways of tackling the barriers to ARV adherence and, therefore, the well-being of people living with HIV. Second, whether focusing on improvements in the quality of life of people living with HIV is more useful than measuring ARV adherence. The study has shown that empowering postpartum women living with HIV as co-researchers to investigate

the barriers to ARV adherence has led to them finding that these barriers are the same as those impeding their quality of life. The co-researchers have, therefore, designed programme interventions that improve the economic, social and psychological well-being of women living with HIV. They believe that improvements in these areas will bring about the well-being of those living with HIV. It is assumed that improvement in ARV adherence would be one of the outcomes of that approach but not the central aim or focus to be measured. Taken together, these findings suggest that focussing on measures of ARV adherence as an indication of the long term well-being of a person living with HIV is flawed as are interventions aimed at only improving rates of ARV adherence. The findings from this study contribute to Pantelic *et al.*'s call for person-centred research and programme development where outcomes are measured by what matters to people living with HIV. The study has provided an example of how postpartum women living with HIV have used Participatory Action Research alongside community-led programme development to effectively target improvements in the lives of people living with HIV. The limitations of this study are that it focussed only on a small number of women, and only women who were postpartum and living in one small area of South Africa. Despite its limitations, the study certainly adds to the understanding and value of participatory, collaborative research and programme development and provides a model for other agencies such as NGOs, health services and academic research initiatives to consider. Significantly more work needs to be done to mainstream this approach. More studies of PAR and community-led development need to be completed and most importantly documented to bring pressure on these agencies to change their modus operandi.

## Supporting information

**S1 File.**
(ZIP)

**S2 File.**
(ZIP)

**S3 File.**
(ZIP)

**S4 File.**
(ZIP)

**S5 File.**
(ZIP)

**S6 File.**
(ZIP)

**S7 File.**
(DOC)

## Acknowledgments

This Photovoice study was submitted in fulfilment of a PhD with the College of Health Sciences at the University of Kwa-Zulu Natal. Staff from the Ubunye Foundation assisted with data collection, analysis and dissemination of findings. Professor Myra Taylor supervised the PhD and contributed to the article text. David and Sue Pepper edited the text and commented on the overall document direction. The most significant contributors to this research study were the 10 women research members and the women from the Lelethu Programme. These

women informed, guided and led the study and continue to make changes in the lives of women living with HIV.

## Author Contributions

**Conceptualization:** Katy Pepper.

**Data curation:** Katy Pepper.

**Formal analysis:** Katy Pepper.

**Funding acquisition:** Katy Pepper.

**Investigation:** Katy Pepper.

**Methodology:** Katy Pepper.

**Project administration:** Katy Pepper.

**Resources:** Katy Pepper.

**Validation:** Katy Pepper.

**Visualization:** Katy Pepper.

**Writing – original draft:** Katy Pepper.

**Writing – review & editing:** Katy Pepper.

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
