## [Decision Letter · Decision Letter 0]

1 Apr 2022

PONE-D-21-37540Replacing sticking plasters with sustainable change: An investigation in an inclusive, community-led approach to addressing the barriers to HIV treatment adherence by people living with HIVPLOS ONE

Dear Dr. Pepper-

Thank you for submitting your manuscript to PLOS ONE. After careful consideration, we feel that it has merit but does not fully meet PLOS ONE’s publication criteria as it currently stands. Therefore, we invite you to submit a revised version of the manuscript that addresses the points raised during the review process.

This paper has received excellent reviews and feedback from a number of the individuals providing the reviews, please adapt the paper appropriately and address the key considerations/revisions suggested to resubmit for reconsideration.  Please submit your revised manuscript by 31 May 2022. If you will need more time than this to complete your revisions, please reply to this message or contact the journal office at plosone@plos.org. Please include the following items when submitting your revised manuscript:A rebuttal letter that responds to each point raised by the academic editor and reviewer(s). You should upload this letter as a separate file labeled 'Response to Reviewers'.A marked-up copy of your manuscript that highlights changes made to the original version. You should upload this as a separate file labeled 'Revised Manuscript with Track Changes'.An unmarked version of your revised paper without tracked changes. You should upload this as a separate file labeled 'Manuscript'.

We look forward to receiving your revised manuscript.

Kind regards,

Melissa Sharer

Academic Editor

PLOS ONE

Journal Requirements:

Additional Editor Comments (if provided):

This paper has received excellent reviews and feedback from a number of the individuals providing the reviews, please adapt the paper appropriately and address the key considerations/revisions suggested.

Reviewers' comments:

Reviewer's Responses to Questions

**Comments to the Author**

1. Is the manuscript technically sound, and do the data support the conclusions?

Reviewer #1: Yes

Reviewer #2: Yes

Reviewer #3: Yes

Reviewer #4: No

2. Has the statistical analysis been performed appropriately and rigorously? 

Reviewer #1: N/A

Reviewer #2: N/A

Reviewer #3: N/A

Reviewer #4: No

3. Have the authors made all data underlying the findings in their manuscript fully available?

Reviewer #1: Yes

Reviewer #2: Yes

Reviewer #3: No

Reviewer #4: No

4. Is the manuscript presented in an intelligible fashion and written in standard English?

Reviewer #1: Yes

Reviewer #2: Yes

Reviewer #3: Yes

Reviewer #4: No

5. Review Comments to the Author

Reviewer #1: This paper addresses an important topic: the intersection of HIV adherence and stigma among a highly vulnerable population, namely, postpartum women. The study thus has very high public health significance. Additionally, the use of participatory action research (PAR) and the active, seemingly equitable manner in which study participants were involved in the research is important and offers valuable insights into different, less hierarchical methods of knowledge creation. These are key strengths that make me excited about the paper. Despite this, however, there are many limitations with the current version of the paper, which must be attended to before the paper is suitable for publication in a high-impact journal like PLOS ONE. I discuss my concerns in some detail below:

1. Title: I do not understand the first part of the title "Replacing sticking plastics with sustainable change". What does this mean? I do not see the relevance to the issues discussed in the paper. If it means something important to the study community then this should be explained somewhere in the paper Additionally, I think that the authors should refer explicitly to "postpartum women living with HIV" rather than simply to "people living with HIV" in the title.

2. Methods: the authors should provide more information on the recruitment of the 10 women who participated in the study. How many women were eligible and how were these 10 women subsequently selected? One assumes that there were more than 10 women who fit the inclusion criteria? Additionally, on Table 1 can the authors include the years that study participants had been living with HIV or the year they first learned their hIV status. This is an important variable, given women's concerns about infecting their children which emerges in the findings section. Table 2 can also be incorporated into Table 1 and does not need to stand alone. Alternatively, the Ubunye Foundation team's socio-demographics can simply be described in narrative form, not via a Table. Given that the study is located in South Africa, I believe that the race of the Ubunye researchers should be stated as this could influence the interaction between the co-researchers and Ubunye researchers.

3. Findings: the authors present very interesting study findings but the organization of these findings needs major revision.

a. Despite the use of sub-themes, the findings sometimes read like a program report rather than findings for an academic paper. This is especially the case from pages 21-25. There seems to be a conflation of the key components of the program and study findings. The following sub-headings are not necessary and reinforce the "report"-like nature of the findings: "A request for Lelethu members to meet" (page 24), "September 2021 Review" (page 22), "VideoVoice Letters" (page 23) and "Community-led Programme" (page 21).

b. Additionally, the authors focus on "two main barriers" to taking ARVS (page 14), but do not state what these are. I assumed that the two barriers were "anticipated stigma" and "poverty". If this is the case, it is not clear what the data presented from page 19, under "The key differences for postpartum women" refer to. Are they still part of the barrier "poverty" or "anticipated stigma?"

c. If a sub-heading only has one or two quotes or a couple of sentences then it should be deleted because it suggests that there is insufficient data on the topic. This applies to the data presented from page 24 "A request for Lelethu members to meet" to page 25. All the subheadings used in this section have insufficient data and thus must be collapsed into one paragraph or more data must be presented on each section.

d. Is the picture shown on page 15 taken from a book or is it a drawing made by one of the women?

e. Figure 3 is not necessary in its current form.

4. Discussion: I wonder if the discussion can go beyond summarizing the findings and linking them to other studies. A more critical discussion of the "so what" of the study findings is important. Perhaps an explicit framing of the findings around the concept of "intersectionality" by Crenshaw, might help elevate the study findings to be more critical versus purely descriptive. It appears to me that what the authors are trying to show is that postpartum women face additional challenges that center around their role as mothers, and having HIV is seen as a type of "failing" to be a good mother because of the HIV risk these mothers pose to the children. Notions of "good" mothers vs "bad" mothers are thus central to how these women experience living with HIV. Poverty interferes with these women's abilities to be good mothers. The authors do not present any data on breastfeeding: I would assume that this would be a key concern women have as well. Did they breastfeed their children and worry that this may also have exposed the kids to potential infection? Were any of the women's children living with HIV? This data should be reported in the findings section as it helps shed important insights into "mothering" and "motherhood".

5. References: There are way too many references provided: 56 to be exact. This could partly be due to the fact that the data are from a doctoral thesis. Please streamline your reference section and only site those sources that you refer to directly in the paper.

6. Overall writing: the authors have a tendency to use short paragraphs. Please combine paragraphs for better flow and coherence. For instance, the last paragraph on page 4 could be combined with the first and second paragraphs on page 5. As a rule of thumb, a paragraph that is only two sentences long should not be a stand-alone paragraph. Please review entire manuscript and use longer paragraphs.

Reviewer #2: Thank you for the opportunity to review this manuscript. I found this to be a singular manuscript. It is notable in its use of PAR. While the findings are not new, certainly the use of PAR with the priority population is new. I was impressed with the engagement of 10 community women, who appear to be extremely vulnerable, as co-researchers. The figures are very helpful. The manuscript held my attention and I feel that it makes a unique contribution.

With all that said, there are some minor revisions that I feel need to be made.

The term "sticking plaster" was unclear to me. The term is used in the title and in the conclusion, and I think that the term is similar to the American version of "putting a bandaid" on a problem. However, it would be helpful to the reader if the authors could clarify the term. Also, did use of the term "sticking plaster" come from data analysis? the co-researchers?

There are some grammatical and language issues that need to be addressed - for example a number of sentences start with "they" and I had to re-read the sentences to better understood who/what "they" meant. I suggest revising to clarify by using nouns, rather than pronouns, to start sentences.

There are a number of what appear to be one sentence paragraphs - - perhaps this is just a formatting issue.

Reviewer #3: Overall, the article is very interesting. I haven't seen nor read a similar studies before, which mean the novelty is good. The modification of photovoice method, to protect the anonymity of PLWHA, was commendable from an ethical point-of-view, and I hope that method would be replicable for similar studies involving vulnerable population. However, I suggest the writing to be more concise and straightforward in the method section, with less focus on the old photovoice model and more on the modified one. I also did not see a clear timeline of when the data was collected.

Further I read into result and discussion section, I got the impression that there were at least 2 studies in this article. One using Photovoice and one using Video Letters as an evaluation of Lelethu program (?). While the notion of inclusivity and participatory research that lead to community-led program to reduce perceived-barriers to ARV-adherence is admirable, this could be at least two different scientific articles with more deep and focused discussion section. My suggestion is the author may chose to focus on one, either the one with photovoice method or the evaluation of Lelethu (?) program.

Reviewer #4: The topic is interesting and has potential to inform how researchers work with people living with HIV and organizations that serve them. However, as currently written, the manuscript requires more clarity about its objectives, clear alignment of the manuscript’s content with its objectives, and better organization of multiple sections of the manuscript. The presentation of findings, the introduction, description of the methods, and results were disorganized and confusing to follow. The program development steps based on qualitative data findings were unclear/link not compellingly described. For example, it is unclear how the two important barriers to ARV adherence (anticipated stigma and poverty) were addressed in the program development.

I offer the following suggestions for the author to consider.

1. Study title: It is not obvious to me how the first part of the title (“replacing sticking plasters …”) reflects the content of the article.

2. Throughout the manuscript, the author described the inclusive and community-led aspects of the project. Yet, the manuscript has one author. Please clarify the role of the co-researchers and the CBO in disseminating and writing-up study results.

3. Introduction: Overall, the introduction reads well and provides relevant info to set up the rest of the manuscript. Organization can be improved. Details are below.

a. Page 4: Please clarify what you meant by measures. If referring to programs or interventions, please use these terms. Measures could be confused with instruments/scales etc.

b. Page 5: Please include relevant stats, e.g., data comparing poverty levels among female-headed households by HIV status.

c. Page 6: It seems poverty and low socioeconomic status are being used interchangeably. I would acknowledge the identical use of these terms, while also noting limitations when researchers use poverty and low socioeconomic status interchangeably.

d. Page 6: Did the review only identify anticipated stigma as a barrier to treatment initiation? How about other stigma experiences, e.g., perceived, internalized and enacted stigmas? Also, please define anticipated HIV stigma.

e. Page 7, 1st sentence, 2nd para. “This study was developed to enable postpartum women to unpack and begin to address these pressures.” Please specify “these pressures.” Are they stigma and poverty?

f. Page 7, 2nd para beginning 2nd sentence should be moved to the methods section.

g. Page 8 the entire page should be moved to the methods section.

h. Please acknowledge the previous studies that used photovoice methods when working with people living with HIV and situate your study given previous similar works. Please refer to: Teti M, Koegler E, Conserve DF, Handler L, Bedford M. A Scoping Review of Photovoice Research Among People With HIV. J Assoc Nurses AIDS Care. 2018 Jul-Aug;29(4):504-527. doi: 10.1016/j.jana.2018.02.010.

4. Methods:

a. Page 9: Please clarify if ethics approval was obtained for the entire project or the PhotoVoice research component only.

b. Page 9, 2nd para: Please revise that information and consent forms do not only allow recordings of discussions and photographs; more importantly, they are included to protect research participants' welfare and ensure that their participation is informed and voluntary.

c. Page 9, 3rd para: Please provide examples of how this process was done by the participants. Were they allowed to show their faces, other peoples' faces?

d. Page 10, 2nd para: How were those women selected by the clinic staff? In addition to health conditions, were there other inclusion criteria?

e. Page 10, 2nd para: Please elaborate on the role of the women co-researchers, including any training they received related to PhotoVoice (e.g., topics/themes that their photos should capture, selection of pictures, confidentiality and safety, number of photos taken). How did the women co-researchers select the photos to be presented?

f. Page 12: Rather than a general description of what are PAR and the PhotoVoice methodology, it would be more instructive to describe how these methods were used in the current study.

g. Page 13: Study procedures generally include all the research procedures conducted to obtain the project's goals, including sampling, design, data collection, etc.

h. Page 13: “Changes to the PhotoVoice process” should be combined with the earlier section on PAR using PhotoVoice methodology.

i. Page 13: “Data analysis at both PhotoVoice and Programme M&E” is more about data management. Please elaborate more on the qualitative analysis, including specific methods or techniques, used to analyze the data, including the role of women co-researchers and Ubunye Foundation.

j. Page 14: “Programme development” does not describe the process of using data to develop a program. Please be more specific about how the data collected were used to inform program development.

k. Figure 2. “Steps for programme development” should be a section on its own in the results section. A figure that illustrates the various steps is helpful, but not enough, given the importance of programme development in the entire project. I would suggest creating a section on programme development that describes in greater detail how the findings from the PhotoVoice activity were used to develop the program.

5. Results

a. Page 19: “The key differences for post-partum women.” Please clarify this section’s purpose. How are the results described here different from the earlier findings about poverty and anticipated stigma?

b. Page 20: “The impact of health facility responses” The title is misleading, as the section does not describe any impact that could be attributed to health facility responses.

c. Page 21: In the preceding page, the author spoke about program development informed by study findings. But here, the subsection is title community-led program: monitoring and evaluation. Before discussing the M&E, please include your findings supporting the development of program that was implemented and evaluated. Again, please discuss how the qualitative research component of the study informed the program development.

d. Page 22: “September 2021 Review” This section does not fit here. The results section could also be better organized and clearer.

e. Page 23: “VideoVoice Letters.” What were the VideoVoice letters and their objective(s)? How were they originally used in the study? This appears to be the first time these letters were mentioned. Please include a description of VideoVoice letters in your methods.

f. Pages 23–25: “Anticipated stigma” This entire subsection on managing anticipated stigma is confusing, and I am not sure what's the purpose for including a separate subsection on "managing anticipated stigma."

6. PLOS authors have the option to publish the peer review history of their article (what does this mean?). If published, this will include your full peer review and any attached files.

Reviewer #1: No

Reviewer #2: No

Reviewer #3: No

Reviewer #4: **Yes: **Rainier Masa

---

## [Author Response · Author response to Decision Letter 0]

15 May 2022

I have responded in detail to all four reviewers comments and suggestions in the requested rebuttal letter which is attached in the document labelled Responses to Reviewers.

---

## [Decision Letter · Decision Letter 1]

28 Jun 2022

“ If I don’t take my treatment, I will die and who will take care of my child ?”: An investigation into an inclusive community-led approach to addressing the barriers to HIV treatment adherence by postpartum women living with HIV

PONE-D-21-37540R1

Dear Dr. Pepper,

We’re pleased to inform you that your manuscript has been judged scientifically suitable for publication and will be formally accepted for publication once it meets all outstanding technical requirements.

Kind regards,

Hugh Cowley

Staff Editor

PLOS ONE

Additional Editor Comments (optional):

Please note that Reviewer #3 has advised that the Methods section of your manuscript should be more technical and concise, and to give a systematic explanation of the study process. Our assessment is that the substance of your Methods section has been sufficiently addressed in your response to multiple reviewers' comments, and that these comments are focused more on the style of presentation. However, if you would like to make any changes to your manuscript in response to the comments below, you will be able to do so as part of the production process.

Reviewers' comments:

Reviewer's Responses to Questions

**Comments to the Author**

1. If the authors have adequately addressed your comments raised in a previous round of review and you feel that this manuscript is now acceptable for publication, you may indicate that here to bypass the “Comments to the Author” section, enter your conflict of interest statement in the “Confidential to Editor” section, and submit your "Accept" recommendation.

Reviewer #1: All comments have been addressed

Reviewer #3: (No Response)

Reviewer #4: All comments have been addressed

2. Is the manuscript technically sound, and do the data support the conclusions?

Reviewer #1: Yes

Reviewer #3: Partly

Reviewer #4: Yes

3. Has the statistical analysis been performed appropriately and rigorously? 

Reviewer #1: N/A

Reviewer #3: N/A

Reviewer #4: Yes

4. Have the authors made all data underlying the findings in their manuscript fully available?

Reviewer #1: Yes

Reviewer #3: Yes

Reviewer #4: Yes

5. Is the manuscript presented in an intelligible fashion and written in standard English?

Reviewer #1: Yes

Reviewer #3: Yes

Reviewer #4: Yes

6. Review Comments to the Author

Reviewer #1: Thank you for the extensive revisions you made to the paper. This revised version is substantially improved. The authors addressed all my concerns and they did a great job at revising the results section so that it flows better and is not choppy nor report-like. There is a a clear and logical connection between the different data sources (e.g., memos and photos) and the subheadings used make sense. The authors also clarified their participant recruitment and selection methods and better explained certain aspects of the paper/population that were not clear to me initially, such as when women were diagnosed with HIV etc. The authors were also very responsive to the other reviewers' comments. Well done. I am happy with this version and recommend that it is accepted for publication.

Reviewer #3: The method section should be more technical and concise, with systematic explanation of the study process (from recruitment, data collection, analysis, etc).

Reviewer #4: Thank you for addressing all comments and responding clearly to questions. I have no additional comments.

7. PLOS authors have the option to publish the peer review history of their article (what does this mean?). If published, this will include your full peer review and any attached files.

Reviewer #1: **Yes: **Tsitsi B Masvawure

Reviewer #3: No

Reviewer #4: **Yes: **Rainier Masa
